# BYPASSING LOGITS BIAS IN ONLINE CLASS-INCREMENTAL LEARNING WITH A GENERATIVE FRAMEWORK

## ABSTRACT

Continual learning requires the model to maintain the learned knowledge while learning from a non-i.i.d data stream continually. Due to the single-pass training setting, online continual learning is very challenging, but it is closer to the real-world scenarios where quick adaptation to new data is appealing. In this paper, we focus on *online class-incremental learning* setting in which new classes emerge over time. Almost all existing methods are replay-based with a softmax classifier. However, the inherent *logits bias* problem in the softmax classifier is a main cause of catastrophic forgetting while existing solutions are not applicable for online settings. To bypass this problem, we abandon the softmax classifier and propose a novel generative framework based on the feature space. In our framework, a generative classifier which utilizes replay memory is used for inference, and the training objective is a pair-based metric learning loss which is proven theoretically to optimize the feature space in a generative way. In order to improve the ability to learn new data, we further propose a hybrid of generative and discriminative loss to train the model. Extensive experiments on several benchmarks, including newly introduced task-free datasets, show that our method beats a series of state-of-the-art replay-based methods with discriminative classifiers, and reduces catastrophic forgetting consistently with a remarkable margin.

## 1 INTRODUCTION

Humans excel at continually learning new skills and accumulating knowledge throughout their lifespan. However, when learning a sequential of tasks emerging over time, neural networks notoriously suffer from *catastrophic forgetting* (McCloskey & Cohen, 1989) on old knowledge. This problem results from non-i.i.d distribution of data streams in such a scenario. To this end, *continual learning* (CL) (Parisi et al., 2019; Lange et al., 2019) has been proposed to bridge the above gap between intelligent agents and humans.

In common CL settings, there are clear boundaries between distinct tasks which are known during training. Within each task, a batch of data are accumulated and the model can be trained offline with the i.i.d data. Recently, online CL (Aljundi et al., 2019c;a) setting has received growing attention in which the model needs to learn from a non-i.i.d data stream in online settings. At each iteration, new data are fed into the model only once and then discarded. In this manner, task boundary is not informed, and thus online CL is compatible with task-free (Aljundi et al., 2019b; Lee et al., 2020) scenario. In real-world scenarios, the distribution of data stream changes over time gradually instead of switching between tasks suddenly. Moreover, the model is expected to quickly adapt to large amount of new data, e.g. user-generated content. Online CL meets these requirements, so it is more meaningful for practical applications. Many existing CL works deal with task-incremental learning (TIL) setting (Kirkpatrick et al., 2017; Li & Hoiem, 2018), in which task identity is informed during test and the model only needs to classify within a particular task. However, for online CL problem, TIL is not realistic because of the dependence on task boundary as discussed above and reduces the difficulty of online CL. In contrast, *class-incremental learning* (CIL) setting (Rebuffi et al., 2017) requires the model to learn new classes continually over time and classify samples over all seen classes during test. Thus, *online CIL* setting is more suitable for online data streams in real-world CL scenarios (Mai et al., 2021).

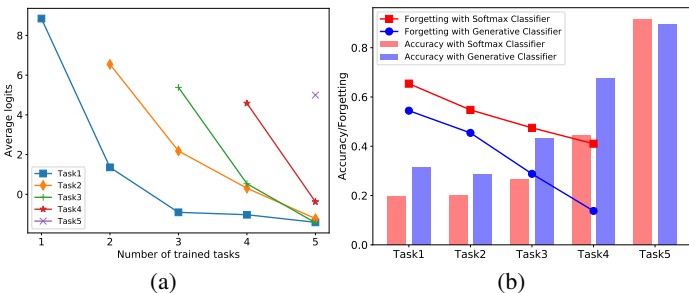

(a)  (b)

Figure 1: Logits bias phenomenon of softmax classifier (left) and accuracy & forgetting on different tasks using softmax vs. generative NCM classifier (right). The results are obtained with ER and 1k replay memory on 5-task Split CIFAR10.

Most existing online CIL methods are based on experience replay (ER) (Robins, 1995; Riemer et al., 2019) strategy which stores a subset of learned data in a replay memory and uses the data in memory to retrain model thus alleviating forgetting. Recently, in CIL setting *logits bias* problem in the last fully connected (FC) layer, i.e. softmax classifier, is revealed (Wu et al., 2019), which is a main cause of catastrophic forgetting. In Figure 1a, we show in online CIL, even if ER is used, logits bias towards newly learned classes in the softamx classifier is still serious and the forgetting on old tasks is dramatic (See Figure 1b). Although some works (Wu et al., 2019; Belouadah & Popescu, 2019; Zhao et al., 2020) propose different methods to reduce logits bias, they all depend on task boundaries and extra offline phases during training so that not applicable for online CIL setting.

In this paper, we propose to tackle the online CIL problem without the softmax classifier to avoid logits bias problem. Instead, we propose a new framework where training and inference are both in a generative way. We are motivated by the insight that generative classifier is more effective in low data regime than discriminative classifier which is demonstrated by Ng & Jordan (2001). Although the conclusion is drawn on simple linear models (Ng & Jordan, 2001), similar results are also observed on deep neural networks (DNNs) (Yogatama et al., 2017; Ding et al., 2020) recently. It should be noticed that in online CIL setting the data is seen only once, not fully trained, so it is analogous to the low data regime in which the generative classifier is preferable. In contrast, the commonly used softmax classifier is a discriminative model.

Concretely, we abandon the softmax FC layer and introduce *nearest-class-mean* (NCM) classifier (Mensink et al., 2013) for inference, which can be interpreted as classifying in a generative way. The NCM classifier is built on the feature space on the top of previous network layers. Thanks to ER strategy, NCM classifier can utilize the replay memory for inference. As for training, inspired by a recent work (Boudiaf et al., 2020), which shows pair-based deep metric learning (DML) losses can be interpreted as optimizing the feature space from a generative perspective, we introduce Multi-Similarity (MS) loss (Wang et al., 2019) to obtain a good feature space for NCM classifier. Meanwhile, we prove theoretically that MS loss is an alternative to a training objective of the generative classifier. In this way, we can bypass logits bias.

To strengthen the model's capable of learning from new data in complex data streams, we further introduce an auxiliary proxy-based DML loss (Movshovitz-Attias et al., 2017). Therefore, our whole training objective is a hybrid of generative and discriminative losses. During inference, we ignore the discriminative objective and classify with the generative NCM classifier. By tuning weight of the auxiliary loss, our method can work well in different data streams.

In summary, our *contributions* are as follows:

1. We make the first attempt to avoid logits bias problem in online CIL setting. In our generative framework, a generative classifier is introduced to replace softmax classifier for inference and for training, we introduce MS loss which is proven theoretically to optimize the model in a generative way.

2. In order to improve the ability of MS loss to learn from new data, we further introduce an auxiliary loss to achieve a good balance between retaining old knowledge and learning new knowledge.

3. We conduct extensive experiments on four benchmarks in multiple online CIL settings, including a new task-free setting we design for simulating more realistic scenarios. Empirical results demonstrate our method outperforms a variety of state-of-the-art replay-based methods substantially, especially alleviating catastrophic forgetting significantly.

## 2 RELATED WORK

Current CL methods can be roughly divided into three categories: which are regularization, parameter isolation and replay-based respectively (Lange et al., 2019). *Regularization* methods retain the learned knowledge by imposing penalty constraints on model's parameters (Kirkpatrick et al., 2017) or outputs (Li & Hoiem, 2018) when learning new data. They work well in TIL setting but poor in CIL setting (van de Ven & Tolias, 2018). *Parameter isolation* methods assign a specific subset of model parameters, such as network weights (Mallya & Lazebnik, 2018) and sub-networks (Fernando et al., 2017) to each task to avoid knowledge interference and thus the network may keep growing. This type of method is mainly designed for TIL as task identity is usually necessary during test. The mainstream of *Replay-based* methods is ER-like (Rebuffi et al., 2017), which stores a subset of old data and retrains it when learning new data to prevent forgetting of old knowledge. In addition, generative replay method trains a generator to replay old data approximately (Shin et al., 2017).

In the field of online CL, most of methods are on the basis of ER. Chaudhry et al. (2019b) first explored ER in online CL settings with different memory update strategies. Authors suggested ER method should be regarded as an important baseline as in this setting it is more effective than several existing CL methods, such as A-GEM (Chaudhry et al., 2019a). GSS (Aljundi et al., 2019c) designs a new memory update strategy by encouraging the divergence of gradients of samples in memory. MIR (Aljundi et al., 2019a) is proposed to select the maximally interfered samples from memory for replay. GMED (Jin et al., 2020) edits the replay samples with gradient information to obtain samples likely to be forgotten, which can benefit the replay in the future. Mai et al. (2021) focus on online CIL setting and adopt the notion of Shapley Value to improve the replay memory update and sampling. All of the above methods are replay-based with softmax classifier. A contemporary work (Lange & Tuytelaars, 2020) proposes CoPE, which is somewhat similar to our method. CoPE replaces softmax classifier with a prototype-based classifier which is non-parametric and updated using features of data samples. However, the loss function of CoPE is still discriminative and the way to classify is analogous to softmax classifier.

Apart from the above ER based methods, Zeno et al. (2018) propose an online regularization method, however it performs very badly in online CL settings (Jin et al., 2020). Lee et al. (2020) propose a parameter isolation method in which the network is dynamically expanded and a memory for storing data is still required. Therefore, the memory usage is not fixed and potentially unbounded.

A recent work (Yu et al., 2020) proposes SDC, a CIL method based on DML and NCM classifier. However, SDC requires an extra phase to correct semantic drift after training each task. This phase depends on task boundaries and the accumulated data of a task, which is not applicable for online CIL. In contrast, our method is based on ER and classifies with replay memory and thus need not correct the drift.

## 3 ONLINE CLASS-INCREMENTAL LEARNING WITH A GENERATIVE FRAMEWORK

### 3.1 PRELIMINARIES AND MOTIVATIONS

#### 3.1.1 ONLINE CLASS-INCREMENTAL LEARNING

CIL setting has been widely used in online CL literature, e.g. (Aljundi et al., 2019a; Mai et al., 2021), and a softmax classifier is commonly used. A neural network $f(\cdot; \boldsymbol{\theta}) : \mathcal{X} \to \mathbb{R}^d$ parameterized by $\boldsymbol{\theta}$ encodes data samples $\boldsymbol{x} \in \mathcal{X}$ into a $d$-dimension feature $f(\boldsymbol{x})$ on which an FC layer $g$ outputs logits for classification: $\boldsymbol{o} = \mathbf{W} f(\boldsymbol{x}) + \mathbf{b}$. At each iteration, a minibatch of data $\mathcal{B}_n$ from a data stream $\mathcal{S}$

arrives and the whole model $(f, g)$ is trained on $\mathcal{B}_n$ only once. The training objective is cross-entropy (CE) loss:

$$\mathcal{L}_{CE} = -\sum_{c=1}^{\tilde{C}} \boldsymbol{t}_{:c} \log \hat{\boldsymbol{y}}_{:c}, \quad \hat{\boldsymbol{y}}_{:c} = \frac{e^{\boldsymbol{o}_{:c}}}{\sum_{c=1}^{\tilde{C}} e^{\boldsymbol{o}_{:c}}} \tag{1}$$

where $\tilde{C}$ is the number of classes seen so far. $\boldsymbol{t}$ is one-hot label of $\boldsymbol{x}$ and the subscript $:c$ denotes the $c$-th component. The new classes from $\mathcal{S}$ emerge over time. The output space of $g$ is the number of seen classes and thus keeps growing. At test time, the model should classify over all $C$ classes seen.

### 3.1.2 EXPERIENCE REPLAY FOR ONLINE CONTINUAL LEARNING

ER makes two modifications during online training: (1) It maintains a replay memory $\mathcal{M}$ with limited size which stores a subset of previously learned samples. (2) When a minibatch of new data $\mathcal{B}_n$ is coming, it samples a minibatch $\mathcal{B}_r$ from $\mathcal{M}$ and uses $\mathcal{B}_n \cup \mathcal{B}_r$ to optimize the model with one SGD-like step. Then it updates $\mathcal{M}$ with $\mathcal{B}_n$. Recent works, e.g. (Aljundi et al., 2019a; Mai et al., 2021) regard ER-reservoir as a strong baseline, which combines ER with reservoir sampling (Vitter, 1985) for memory update and random sampling for $\mathcal{B}_r$. See Chaudhry et al. (2019b) for more details about it.

### 3.1.3 LOGITS BIAS IN SOFTMAX CLASSIFIER

Some recent works (Wu et al., 2019; Belouadah & Popescu, 2019; Zhao et al., 2020) show in CIL scenarios, even with replay-based mechanism the logits outputted by model always have a strong bias towards the newly learned classes, which leads to catastrophic forgetting actually. In preliminary experiments, we also observe this phenomenon in online CIL setting. We run ER-reservoir baseline on 5-task Split CIFAR10 (each task has two disjoint classes) online CIL benchmark. In Figure 1a, we display the average logits of each already learned tasks over samples in test data after learning each task. The model outputs much higher logits on the new classes (of the task just learned) than old classes.

Following (Ahn & Moon, 2020), we examine the CE loss in Eq (1), the gradient of $\mathcal{L}_{CE}$ w.r.t logit $\boldsymbol{o}_{:c}$ of class $c$ is $\hat{\boldsymbol{y}}_{:c} - \mathbb{I}[\boldsymbol{t}_{:c} = 1]$. Thus, if $c$ is the real label $y$, i.e. $\boldsymbol{t}_{:c} = 1$, the gradient is non-positive and model is trained to increase $\boldsymbol{o}_{:c}$, otherwise the gradient is non-negative and model is trained to decrease $\boldsymbol{o}_{:c}$. Therefore, *logits bias* problem is caused by the imbalance between the number of samples of the new classes and that of the old classes with a limited size of $\mathcal{M}$. As mentioned in Section 1, existing solutions (Wu et al., 2019; Belouadah & Popescu, 2019; Zhao et al., 2020) designed for conventional CIL need task boundaries to conduct extra offline training phases and even depend on the accumulated data of one task. They are not applicable for online CIL setting where task boundaries are not informed or even do not exist in task-free scenario.

## 3.2 INFERENCE WITH A GENERATIVE CLASSIFIER

Proposed generative framework is based on ER strategy, and aims to avoid the intrinsic logits bias problem by removing the softmax FC layer $g$ and build a generative classifier on the feature space $\mathcal{Z} : \boldsymbol{z} = f(\boldsymbol{x})$. If the feature $\boldsymbol{z}$ is well discriminative, we can conduct inference with samples in $\mathcal{M}$ instead of a parametric classifier which is prone to catastrophic forgetting (Rebuffi et al., 2017). We use NCM classifier firstly suggested by Rebuffi et al. (2017) for CL and show it is a generative model. We use $\mathcal{M}_c$ to denote the subset of class $c$ of $\mathcal{M}$. The class mean $\boldsymbol{\mu}_c$ is computed by $\boldsymbol{\mu}_c^{\mathcal{M}} = \frac{1}{|\mathcal{M}_c|} \sum_{\boldsymbol{x} \in \mathcal{M}_c} f(\boldsymbol{x})$. During inference, the prediction for $\boldsymbol{x}^*$ is made by:

$$y^* = \arg \min_c \|f(\boldsymbol{x}^*) - \boldsymbol{\mu}_c^{\mathcal{M}}\|_2 \tag{2}$$

In fact, the principle of prediction in Eq (2) is to find a Gaussian distribution $\mathcal{N}(f(\boldsymbol{x}^*)|\boldsymbol{\mu}_c^{\mathcal{M}}, \mathbf{I})$ with the maximal probability for $\boldsymbol{x}^*$. Therefore, assuming the conditional distribution $p(\boldsymbol{z}|y = c) = \mathcal{N}(\boldsymbol{z}|\boldsymbol{\mu}_c^{\mathcal{M}}, \mathbf{I})$ and the prior distribution $p(y)$ is uniform, NCM classifier virtually deals with $p(y|\boldsymbol{z})$ by modeling $p(\boldsymbol{z}|y)$ in a generative way. The inference way is according to Bayes rule: $\arg \max p(y|\boldsymbol{x}^*) = \arg \max p(f(\boldsymbol{x}^*)|y)p(y)$. The assumption about $p(y)$ simplifies the analysis

and works well in practice. In contrast, softmax classifier models $p(y|\boldsymbol{x}^*)$ in a typical discriminative way.

As discussed above, online CIL is in a low data setting where generative classifiers are preferable compared to discriminative classifiers (Ng & Jordan, 2001). Moreover, generative classifiers are more robust to continual learning (Yogatama et al., 2017) and imbalanced data settings (Ding et al., 2020). At each iteration, $\mathcal{B}_n \cup \mathcal{B}_r$ is also highly imbalanced. Considering these results, we hypothesis generative classifiers are promising for online CIL problem. It should be noted our method only models a simple generative classifier $p(\boldsymbol{z}|y)$ on the feature space, instead of modeling $p(\boldsymbol{x}|y)$ on the input space using DNNs (Yogatama et al., 2017), which is time-consuming and thus is not suitable for online training.

### 3.3 Training with a Pair-based Metric Learning Loss from a Generative Perpective

To train the feature extractor $f(\boldsymbol{\theta})$ we resort to DML losses which aim to learn a feature space where the distances represent semantic dissimilarities between data samples. From the perspective of mutual information (MI), Boudiaf et al. (2020) theoretically show the equivalence between CE loss and several pair-based DML losses, such as contrast loss (Hadsell et al., 2006) and Multi-Similarity (MS) loss (Wang et al., 2019). The DML losses maximize MI between feature $\boldsymbol{z}$ and label $y$ in a generative way while CE loss in a discriminative way, which motivates us to train $f(\boldsymbol{\theta})$ with a pair-based DML loss to obtain a good feature space $\mathcal{Z}$ for the generative classifier.

Especially, we choose the MS loss as a training objective. MS loss is one of the state-of-the-art methods in the field of DML. Wang et al. (2019) point out pair-based DML losses can be seen as weighting each feature pair in the general pair weighting framework. As MS loss requires the feature $f(\boldsymbol{x})$ to be $\ell_2$-normalized first, from now on, we use $\boldsymbol{z}_i$ to denote the $\ell_2$-normalized feature of $\boldsymbol{x}_i$ and a feature pair is represented in the form of inner product $S_{ij} := \boldsymbol{z}_i^{\mathrm{T}} \boldsymbol{z}_j$. To weight feature pairs better, MS loss is proposed to consider multiple types of similarity. MS loss on a dataset $\mathcal{D}$ is formulated as follows:

$$\mathcal{L}_{MS}(\mathcal{D}) = \frac{1}{|\mathcal{D}|} \sum_{i=1}^{|\mathcal{D}|} \left\{ \frac{1}{\alpha} \log[1 + \sum_{j \in \mathcal{P}_i} e^{-\alpha(S_{ij}-\lambda)}] + \frac{1}{\beta} \log[1 + \sum_{j \in \mathcal{N}_i} e^{\beta(S_{ij}-\lambda)}] \right\} \tag{3}$$

where $\alpha$, $\beta$ and $\lambda$ are hyperparameters and $\mathcal{P}_i$ and $\mathcal{N}_i$ represent the index set of positive and negative samples of $\boldsymbol{x}_i$[1] respectively. MS loss also utilizes the hard mining strategy to filter out too uninformative feature pairs, i.e. too similar positive pairs and too dissimilar negative pairs:

$$\mathcal{P}_i = \{j | S_{ij} < \max_{y_k \neq y_i} S_{ik} + \epsilon\} \quad \mathcal{N}_i = \{j | S_{ij} > \min_{y_k = y_i} S_{ik} - \epsilon\} \tag{4}$$

where $\epsilon$ is another hyperparameter in MS loss. At each iteration we use MS loss on the union of new samples and replay samples $\mathcal{L}_{MS}(\mathcal{B}_n \cup \mathcal{B}_r)$ to train the model. The sampling of $\mathcal{B}_r$ and update of $\mathcal{M}$ are the same as ER-reservoir.

To show the connection between $\mathcal{L}_{MS}$ and the generative classifier in Eq (2), we conduct some theoretical analyses.

**Proposition 1.** *Assume dataset $\mathcal{D} = \{(\boldsymbol{x}_i, y_i)\}_{i=1}^n$ is class-balanced and has $C$ classes each of which has $n_0$ samples. For a generative model $p(\boldsymbol{z}, y)$, assume $p(y)$ actually obeys the uniform distribution and $p(\boldsymbol{z}|y = c) = \mathcal{N}(\boldsymbol{z}|\boldsymbol{\mu}_c^{\mathcal{D}}, \mathbf{I})$ where $\boldsymbol{\mu}_c^{\mathcal{D}} = \frac{1}{n_0} \sum_{i=1}^n \boldsymbol{z}_i \mathbb{I}[y_i = c]$. For MS loss assume hard mining in Eq (4) is not employed. Then we have:*

$$\mathcal{L}_{MS} \overset{c}{\geq} \mathcal{L}_{Gen-Bin} \tag{5}$$

*where $\overset{c}{\geq}$ stands for upper than, up to an additive constant $c$ and $\mathcal{L}_{Gen-Bin}$ is defined in the following:*

$$\mathcal{L}_{Gen-Bin} = -\frac{1}{n} \sum_{i=1}^n \log p(\boldsymbol{z}_i|y = y_i) + \frac{1}{nC} \sum_{i=1}^n \sum_{c=1}^C \log p(\boldsymbol{z}_i|y = c) \tag{6}$$

---

[1]The positive samples have the same labels as $\boldsymbol{x}_i$ while the negative samples have different labels from $\boldsymbol{x}_i$.

The proof of Proposition 1 is in Appendix. Proposition 1 shows $\mathcal{L}_{MS}$ is an upper bound of $\mathcal{L}_{Gen-Bin}$ and thus is an alternative to minimizing $\mathcal{L}_{Gen-Bin}$. The first term of $\mathcal{L}_{Gen-Bin}$ aims to minimize the negative log-likelihood of the class-conditional generative classifier, while the second term maximizes the conditional entropy $H(Z|Y)$ of labels $Y$ and features $Z$. It should be noticed $\mathcal{L}_{Gen-Bin}$ depends on modeling $p(\boldsymbol{z}|y)$. With uniform $p(y)$, classifying using $p(\boldsymbol{z}|y)$ equals to classifying using $p(\boldsymbol{z}, y)$, and $H(Z|Y)$ is equivalent to $H(Z, Y)$, which can be regarded as a regularizer against features collapsing. Thus, $\mathcal{L}_{Gen-Bin}$ actually optimizes the model in a generative way. The assumptions in Proposition 1 are similar with those in Section 3.2 about NCM classifier. The difference lies in that NCM classifier uses $\{\boldsymbol{\mu}_c^{\mathcal{M}}\}$ computed on replay memory $\mathcal{M}$ to approximate $\{\boldsymbol{\mu}_c^{\mathcal{D}}\}$. Therefore, Proposition 1 reveals that MS loss optimizes the feature space in a generative way and it models $p(\boldsymbol{z}|y)$ for classification which is consistent with the NCM classifier.

The real class means $\{\boldsymbol{\mu}_c^{\mathcal{D}}\}$ depend on all training data and change with the update of $f(\boldsymbol{\theta})$ so that are intractable in online settings. MS loss can be efficiently computed as it does not depend on $\{\boldsymbol{\mu}_c^{\mathcal{D}}\}$ thus the model can be trained efficiently. During inference, we use approximate class means $\boldsymbol{\mu}_c^{\mathcal{M}}$ to classify. In Figure 1b, on 5-task Split CIFAR10 benchmark, we empirically show compared to softmax classifier, on old tasks, our method achieves much higher accuracy and much lower forgetting, which implies MS loss is an effective objective to train the model $f$ and class mean $\boldsymbol{\mu}_c^{\mathcal{M}}$ of replay memory is a good approximation of $\boldsymbol{\mu}_c^{\mathcal{D}}$.

With discriminative loss like CE loss, the classifier models a discriminative model $p(y|\boldsymbol{z})$. Therefore, if training with discriminative loss and inference with NCM classifier based on the generative model $p(\boldsymbol{z}|y)$, we can not expect to obtain good results. In the next section, experiments will verify this conjecture. In contrast, the way to train and inference are coincided in proposed generative framework.

## 3.4 A Hybrid Generative/Discriminative Loss

However, when addressing classification tasks, generative classifier has natural weakness, since modeling joint distribution $p(\boldsymbol{x}, y)$ is much tougher than modeling conditional distribution $p(y|\boldsymbol{x})$ for NNs. Moreover, in preliminary experiments, we found if only trained with MS loss, the NCM classifier's performance degenerates as the expected number of classes in $\mathcal{B}_n$ at each iteration increases. This phenomenon is attributed to the inadequate ability to learn from new data, instead of catastrophic forgetting. We speculate because the size of $\mathcal{B}_n$ is always fixed to a small value (e.g. 10) in online CIL settings, the number of positive pairs in $\mathcal{B}_n$ decreases as the expected number of classes increases.

To remedy this problem, we take advantage of discriminative losses for fast adaptation in online setting. To this end, we introduce Proxy-NCA (PNCA) (Movshovitz-Attias et al., 2017), a proxy-based DML loss, as an auxiliary loss. For each class, PNCA loss maintains "proxies" as the real feature to utilize the limited data in a minibatch better, which leads to convergence speed-up compared to pair-based DML losses. Concretely, when a new class $c$ emerges, we assign one trainable proxy $\boldsymbol{p}_c \in \mathbb{R}^d$ to it. PNCA loss is computed as:

$$\mathcal{L}_{PNCA}(\mathcal{D}) = -\frac{1}{|\mathcal{D}|} \sum_{(x,y) \in \mathcal{D}} \log \frac{e^{-\|f(\boldsymbol{x}) - \boldsymbol{p}_y\|_2^2}}{\sum_{c=1}^{\tilde{C}} e^{-\|f(\boldsymbol{x}) - \boldsymbol{p}_c\|_2^2}} \tag{7}$$

Movshovitz-Attias et al. (2017) suggest all proxies have the same norm $N_P$ and all features have the norm $N_F$. The latter satisfies as in MS loss the feature is $\ell_2$-normalized, i.e. $N_F = 1$. We also set $N_P = 1$ by normalizing all proxies after each SGD-like update. In this way, $\mathcal{L}_{PNCA}$ is equivalent to a CE loss with $\ell_2$-normalized row vectors of $\mathbf{W}$ and without bias $\mathbf{b}$, and thus we use PNCA instead of CE loss to keep utilizing the normalized features of MS loss. Our full training objective is a hybrid of generative and discriminative losses:

$$\mathcal{L}_{Hybrid} = \mathcal{L}_{MS} + \gamma \mathcal{L}_{PNCA} \tag{8}$$

where $\gamma$ is a hyperparameter to control the weight of $\mathcal{L}_{PNCA}$. In general, generative classifiers have a smaller variance but higher bias than discriminative classifiers, and using such a hybrid loss can achieve a better bias-variance tradeoff (Bouchard & Triggs, 2004). Thus we think introducing the discriminative loss $\mathcal{L}_{PNCA}$ can reduce the bias of model so that boost the ability to learn from new data.

It should be noticed we train the model with $\mathcal{L}_{Hybrid}(\mathcal{B}_n \cup \mathcal{B}_r)$, while we only use NCM classifier in Eq (2) for inference. In all experiments, we set $\alpha = 2$, $\beta = 50$, $\epsilon = 0.1$ following Wang et al. (2019)

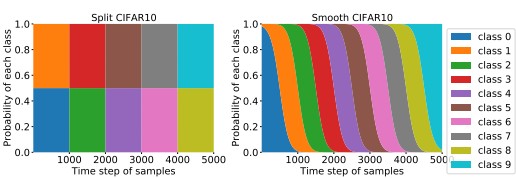
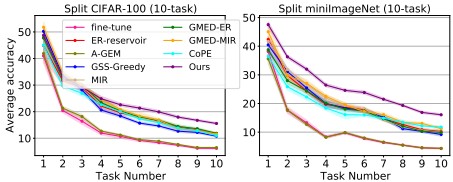

Figure 2: The probability distribution on class of the sample at each time step on Split CIFAR10 (left) and Smooth CIFAR10 (right).

Figure 3: Average Accuracy on already learned task during training.

and $\lambda = 0.5$ which always works well in online CIL settings. We only need to tune hyperparemeters $\gamma$ in Eq (8) for different experiment settings.

# 4 EXPERIMENTS

## 4.1 EXPERIMENT SETUP

**Datasets** First, we conduct experiments on *Split* datasets which are commonly used in CIL and online CIL literature. On Split MNIST and CIFAR10, the datasets are split into 5 tasks each of which comprises 2 classes. On CIFAR100 and miniImageNet with 100 classes, we split them into 10 or 20 tasks. The number of classes in each task is 10 or 5 respectively. For MNIST we select 5k samples for training following Aljundi et al. (2019a) and we use full training data for other datasets. To simulate a task-free scenario, task boundaries are not informed during training (Jin et al., 2020).

To conduct a thorough evaluation in task-free scenarios, we design a new type of data streams. For a data stream with $C$ classes, we assume the length of stream is $n$ and $n_0 = n/C$. We denote $p_c(t)$ as the occurrence probability of class $c$ at time step $t$ and assume $p_c(t) \sim \mathcal{N}(t|(2c-1)n_0/2, n_0/2)$. At each time step $t$, we calculate $\boldsymbol{p}(t) = (p_1(t), \ldots, p_C(t))$ and normalize $\boldsymbol{p}(t)$ as the parameters of a Categorical distribution from which a class index $c_t$ is sampled. Then we sample one data of class $c_t$ without replacement. In this setting, data distribution changes smoothly and there is no notion of task. We call such data streams as *Smooth* datasets. To build Smooth datasets, we set $n = 5k$ on CIFAR10 and $n = 40k$ on CIFAR100 and miniImageNet, using all classes in each dataset. For all datasets, the size of minibatch $\mathcal{B}_n$ is 10. In Figure 2 we plot the probability distribution on class at each time step in the data stream generation process for Split CIFAR10 and Smooth CIFAR10. For better comparison, we set the length of data stream of two datasets both 5k. Split CIFAR10 has clear task boundaries and within one task the distribution on class is unchanged and uniform. However, On Smooth CIFAR10, the distribution on class keeps changing and there is no notion of task.

**Baselines** We compare our method against a series of state-of-the-art online CIL methods, including: ER-reservoir, A-GEM, GSS, MIR, GMED, CoPE and ASER. We have briefly introduced them in Section 2. Specially, we use GSS-greedy and $\text{ASER}_\mu$ which are the best variants in the corresponding paper. For GMED, we evaluate both GMED-ER and GMED-MIR. We also evaluate fine-tune baseline without any CL strategy. For all baselines and our method, the model is trained with 1 epoch, i.e. online CL setting. In addition, the performances of i.i.d online and i.i.d offline are also provided, by training the model 1 and 5 epochs respectively on i.i.d data streams. We reimplement all baselines except $\text{ASER}_\mu$, whose results are from the original paper. **Model** Following Aljundi et al. (2019a), the model $f$ is a 2-layer MLP with 400 hidden units for MNIST and a reduced ResNet18 for other datasets. For baselines with ER strategy, the size of replay minibatch $\mathcal{B}_r$ is always 10. The budget of memory $\mathcal{M}$ is 500 on MNIST and 1000 on others. We use a relatively small budget $|\mathcal{M}|$ to mimic a practical setting. All models are optimized by SGD. The *single-head evaluation* is always used for CIL. More details about datasets, hyperparameter selection and evaluation metrics are in Appendix.

## 4.2 MAIN RESULTS ON *Split* DATASETS

On *Split* datasets, we use **Average Accuracy** and **Average Forgetting** after training all tasks (Chaudhry et al., 2019b) for evaluation, which are reported in Table 1 and Table 2 respectively. For each metric, we report the mean of 15 runs and the 95% confidence interval.

| Methods | MNIST (5-task) | CIFAR10 (5-task) | CIFAR100 (10-task) | CIFAR100 (20-task) | miniImageNet (10-task) | miniImageNet (20-task) |
|---|---|---|---|---|---|---|
| fine-tune | 19.66±0.05 | 18.40±0.17 | 6.26±0.30 | 3.61±0.24 | 4.43±0.19 | 3.12±0.15 |
| ER-reservoir | 82.34±2.48 | 39.88±1.52 | 11.59±0.26 | 8.95±0.26 | 10.24±0.41 | 8.33±0.66 |
| A-GEM | 25.99±1.62 | 18.01±0.17 | 6.48±0.18 | 3.66±0.09 | 4.68±0.11 | 3.37±0.13 |
| GSS-Greedy | 83.88±0.72 | 39.07±2.02 | 10.78±0.28 | 7.94±0.47 | 9.20±0.61 | 7.76±0.35 |
| MIR | 86.81±0.95 | 42.10±1.27 | 11.52±0.37 | 8.61±0.34 | 9.99±0.49 | 7.93±0.70 |
| GMED-ER | 81.71±1.87 | 42.65±1.27 | 11.86±0.36 | 9.16±0.47 | 9.53±0.66 | 8.14±0.58 |
| GMED-MIR | 88.70±0.81 | 44.53±2.23 | 11.58±0.51 | 8.48±0.37 | 9.24±0.53 | 7.75±0.80 |
| CoPE | 87.58±0.65 | 47.36±0.96 | 10.79±0.36 | 9.11±0.44 | 11.03±0.68 | 9.92±0.61 |
| ASER$_\mu$ [*] | – | 43.50±1.40 | 14.00±0.40 | – | 12.20±0.80 | – |
| Ours | **88.79±0.26** | **51.84±0.91** | **15.56±0.39** | **13.65±0.35** | **16.05±0.38** | **15.15±0.36** |
| i.i.d. online | 86.35±0.64 | 62.37±1.36 | 20.62±0.48 | 20.62±0.48 | 18.02±0.63 | 18.02±0.63 |
| i.i.d. offline | 92.44±0.61 | 79.90±0.51 | 45.59±0.29 | 45.59±0.29 | 38.63±0.59 | 38.63±0.59 |

Table 1: Average Accuracy of 15 runs on *Split* datasets. Higher is better. [*] indicates the results are from the original paper.

| Methods | MNIST (5-task) | CIFAR10 (5-task) | CIFAR100 (10-task) | CIFAR100 (20-task) | miniImageNet (10-task) | miniImageNet (20-task) |
|---|---|---|---|---|---|---|
| fine-tune | 99.24±0.09 | 85.45±0.63 | 51.60±0.77 | 65.51±0.78 | 41.12±0.82 | 52.99±0.89 |
| ER-reservoir | 18.33±1.77 | 52.72±1.90 | 45.94±0.55 | 57.31±0.71 | 36.05±0.78 | 47.70±0.90 |
| A-GEM | 89.90±2.02 | 82.80±0.73 | 54.15±0.42 | 67.61±0.53 | 43.31±0.52 | 54.47±0.78 |
| GSS-Greedy | 15.13±0.99 | 49.96±2.82 | 44.30±0.57 | 53.87±0.54 | 36.17±0.58 | 45.91±0.79 |
| MIR | 9.71±1.39 | 44.34±2.65 | 46.52±0.52 | 56.58±0.62 | 36.98±0.78 | 45.84±1.11 |
| GMED-ER | 16.21±2.70 | 44.93±1.68 | 46.35±0.50 | 57.76±0.94 | 35.22±1.16 | 45.08±1.28 |
| GMED-MIR | 12.52±1.05 | 39.88±2.23 | 46.56±0.65 | 58.14±0.55 | 34.79±1.01 | 45.50±1.49 |
| CoPE | 9.51±1.15 | 40.01±1.80 | 36.51±0.86 | 43.82±0.62 | 29.43±0.98 | 40.99±1.02 |
| ASER$_\mu$ [*] | – | 47.90±1.60 | 45.00±0.70 | – | 28.00±1.30 | – |
| Ours | 9.36±0.37 | **35.37±1.35** | **21.79±0.69** | **27.10±1.10** | **21.26±0.59** | **24.98±0.87** |

Table 2: Average Forgetting of 15 runs on *Split* datasets. Lower is better. [*] indicates the results are from the original paper.

In Table 1, we can find our method outperforms all baselines on all 6 settings. The improvement of our method is significant except on MNIST, where GMED-MIR is competitive with our method. An interesting phenomenon is all existing methods do not have a substantial improvement over ER-reservoir on CIFAR100 and miniImageNet, except ASER. We argue for online CIL problem, we should pay more attention to complex settings. Nevertheless, our method is superior to ASER obviously, especially on CIFAR10 and miniImageNet. Table 2 shows the forgetting of our method is far lower than other methods based on the softmax classifier, except on MNIST. Figure 3 shows our method is almost consistently better than all baselines during the whole learning processes. More results with various memory sizes can be found in Appendix.

**Ablation Study** We also conduct ablation study about training objective and classifier in Table 4. The ER-reservoir corresponds to the first row and our method corresponds to the last row. Firstly, we find for ER-reservoir, replacing softmax classifier with NCM classifier makes a substantial improvement on CIFAR10. However, it has no effect on more complex CIFAR100 (row 1&2). Secondly, only using MS loss works very well on CIFAR10 while on CIFAR100 poor ability to learn from new data limits its performance (row 3&6). Lastly, when hybrid loss is used, the NCM classifier is much better than proxy-based classifier (row 5&6), and MS loss is critical for NCM classifier (row 4&6). Note that hybrid loss does not outperform MS loss much on Split-CIFAR10. This is because in Split-CIFAR10, a minibatch of new data contains a maximum of two classes, and thus the positive pairs are enough fo MS loss to learn new knowledge well. These results verify our statement in Section 3.2.

**Comparison with Logits Bias Solutions for Conventional CIL Setting** Although existing CIL methods to alleviate logits bias are not applicable for online CIL settings as task boundaries are necessary, after being modified in some ways they can be adapted to online CIL. To better reflect the contribution of our method, we adapt iCaRL (Rebuffi et al., 2017) and BiC (Wu et al., 2019) to online CIL and compare modified iCaRL and modified BiC with our method in Table 3. iCaRL replaces softmax classifier with NCM classifier and BiC uses a linear bias correction layer to reduce logits bias. and iCaRL is modified in the following way: at each iteration, we minimize binary CE loss used by iCaRL which encourages the model to mimic the outputs for all learned classes of the

| Methods | MNIST (5-task) | CIFAR10 (5-task) | CIFAR100 (20-task) | miniImageNet (20-task) |
|---------|---------|---------|---------|---------|
| modified iCaRL | 34.58 ±1.18 | 29.77 ±0.91 | 5.38 ±0.26 | 7.16 ±0.33 |
| modified BiC | 83.33 ±1.35 | 43.65 ±2.50 | 8.72 ±0.30 | 7.91 ±0.55 |
| Ours | **88.79** **±0.26** | **51.84** **±0.91** | **13.65** **±0.35** | **15.15** **±0.36** |

Table 3: The comparison between modified iCaRL, modified BiC and our method (Ours) on *Split* datasets.

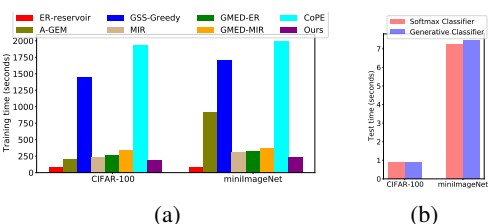

(a)      (b)

Figure 4: Comparison of training time (a) and test time (b) on Split CIFAR100 and miniImageNet. The number of tasks is 10.

| Training | Inference | CIFAR10 | CIFAR100 |
|---------|---------|---------|---------|
| CE loss | Softmax (Dis) | 39.88±1.52 | 8.95±0.26 |
| CE loss | NCM (Gen) | 44.46±0.95 | 8.96±0.38 |
| MS loss | NCM (Gen) | 51.72±1.02 | 9.99±0.32 |
| PNCA loss | NCM (Gen) | 41.91±1.78 | 9.31±0.58 |
| Hybrid loss | Proxy (Dis) | 48.16±1.21 | 7.02±0.75 |
| Hybrid loss | NCM (Gen) | 51.84±0.91 | 13.65±0.35 |

Table 4: Ablation study on Split CIFAR10 and 20-task Split CIFAR100. We show the performances of different combinations of losses and inference ways. Dis: Discriminative, Gen: Generative.

| Method | CIFAR10 | CIFAR100 | miniImageNet |
|---------|---------|---------|---------|
| fine-tune | 10.02±0.03 | 1.02±0.03 | 1.02±0.04 |
| ER-reservoir | 20.89±2.07 | 3.84±0.42 | 6.85±0.70 |
| MIR | 18.75±2.53 | 4.35±0.53 | 6.09±1.04 |
| GMED-MIR | 18.78±2.31 | 3.68±0.48 | 7.22±0.81 |
| Ours | **34.18±0.81** | **10.54±0.38** | **12.24±0.19** |
| i.i.d online | 31.23±2.11 | 18.08±0.62 | 17.23±0.42 |
| i.i.d offline | 48.37±1.23 | 42.68±0.37 | 39.82±0.46 |

Table 5: Final accuracy of 15 runs on *Smooth* datasets.

old model after the last iteration. We use reservoir sampling for memory update. NCM classifier is used for inference. For modified BiC, we use the linear bias correction layer of BiC to correct the logits for all learned classes only before test as task boundaries are unavailable in online CIL setting. As shown in Table 3, modified iCaRL performs very badly and the performances of modified BIC are much worse than our method. These results imply adapting existing methods to online setting can not alleviate logits bias effectively for online CIL. Therefore, our method makes a substantial contribution for online CIL.

**Time Comparison** In Figure 4a, we report the training time of different methods. The training time of our method is only a bit higher than ER-reservoir.

Most baselines, such as GSS, MIR, and GMED, improve ER-reservoir by designing new memory update and sampling strategies which depend on extra gradient computations and thus are time-consuming. The inference costs of softmax classifier and our NCM classifier are displayed in Figure 4b. We can find the extra time of NCM to compute the class means is (about 3%) slight, as the size of memory is limited.

### 4.3 RESULTS ON TASK-FREE *Smooth* DATASETS

In newly designed task-free *Smooth* datasets, the new classes emerge irregularly and the distribution on class changes at each time step. In Table 5, we compare our method with several baselines on three *smooth* datasets. The metric is final accuracy after learning the whole data stream. We can find these datasets are indeed more complex as *fine-tune* can only classify correctly on the last class, which is due to the higher imbalance of data streams. For this reason, baselines such as ER-reservoir and MIR degrade obviously compared with *split datsets*. However, our method performs best consistently.

## 5 CONCLUSION

In this work, we tackle with online CIL problem from a generative perspective to bypass logits bias problem in commonly used softmax classifier. We first propose to replace softmax classifier with a generative classifier. Then we introduce MS loss for training and prove theoretically that it optimizes the feature space in a generative way. We further propose a hybrid loss to boost the model's ability to learn from new data. Experimental results show the significant and consistent superiority of our method compared to existing state-of-the-art methods.

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

# A    PROOFS

## A.1    PROPOSITION 1

*Proof.* For simplicity, we ignore the coefficient $\frac{1}{n}$ both in $\mathcal{L}_{MS}$ and $\mathcal{L}_{Gen-Bin}$ in the proof. We denote the two parts in the summation of $\mathcal{L}_{MS}$ as $\mathcal{L}_{MS}^+$ and $\mathcal{L}_{MS}^-$ respectively, i.e.:

$$
\begin{aligned}
\mathcal{L}_{MS}^+ &= \sum_{i=1}^n \frac{1}{\alpha} \log[1 + \sum_{j \in \mathcal{P}_i} e^{-\alpha(S_{ij} - \lambda)}] \\
\mathcal{L}_{MS}^- &= \sum_{i=1}^n \frac{1}{\beta} \log[1 + \sum_{j \in \mathcal{N}_i} e^{\beta(S_{ij} - \lambda)}]
\end{aligned}
\tag{9}
$$

Without hard mining, for $\mathcal{L}_{MS}^+$ we have:

$$
\begin{aligned}
\mathcal{L}_{MS}^+ &= \sum_{i=1}^n \frac{1}{\alpha} \log[1 + \sum_{j:y_j=y_i} e^{-\alpha(S_{ij} - \lambda)}] \\
&\geq \sum_{i=1}^n \frac{1}{\alpha} \log[\sum_{j:y_j=y_i} e^{-\alpha(S_{ij} - \lambda)}] \\
&\overset{c}{\geq} \sum_{i=1}^n \frac{1}{\alpha}[\frac{1}{n_0} \sum_{j:y_j=y_i} \log e^{-\alpha(S_{ij} - \lambda)}] \\
&\quad \triangleright \text{Jensen's inequality} \\
&= \sum_{i=1}^n \frac{1}{n_0}[\sum_{j:y_j=y_i} -(S_{ij} - \lambda)] \\
&\overset{c}{=} -\sum_{i=1}^n \frac{1}{n_0} \sum_{y_j=y_i} S_{ij}
\end{aligned}
\tag{10}
$$

where $\stackrel{c}{=}$ stands for equal to, up to an additive constant. For $\mathcal{L}_{MS}^{-}$ we can write:

$$
\begin{aligned}
\mathcal{L}_{MS}^{-} &= \sum_{i=1}^{n} \frac{1}{\beta} \log[1 + \sum_{j:y_j \neq y_i} e^{\beta(S_{ij}-\lambda)}] \\
&\geq \sum_{i=1}^{n} \frac{1}{\beta} \log[\sum_{j:y_j \neq y_i} e^{\beta(S_{ij}-\lambda)}] \\
&\stackrel{c}{\geq} \sum_{i=1}^{n} \frac{1}{\beta}[\frac{1}{n_0(C-1)} \sum_{j:y_j \neq y_i} \log e^{\beta(S_{ij}-\lambda)}] \\
&\qquad \triangleright \text{Jensen's inequality} \\
&= \sum_{i=1}^{n} \frac{1}{n_0(C-1)} \sum_{j:y_j \neq y_i} S_{ij}
\end{aligned}
\tag{11}
$$

According to Eq (10) and Eq (11), we have:

$$
\mathcal{L}_{MS} \stackrel{c}{\geq} \sum_{i=1}^{n} \left\{ -\frac{1}{n_0} \sum_{j:y_j=y_i} S_{ij} + \frac{1}{(C-1)n_0} \sum_{j:y_j \neq y_i} S_{ij} \right\}
\tag{12}
$$

Now we consider $\mathcal{L}_{Gen-Bin}$. Firstly, it can be written:

$$
\begin{aligned}
\mathcal{L}_{Gen-Bin} \stackrel{c}{=} \sum_{i=1}^{n} \Big\{ &-\log p(\boldsymbol{z}_i|y=y_i) \\
&+ \frac{1}{C-1} \sum_{c=1,c\neq y_i}^{C} \log p(\boldsymbol{z}_i|y=c) \Big\}
\end{aligned}
\tag{13}
$$

For convenience, we denote the features of data samples whose labels are $c$ as $\{\boldsymbol{z}_{c_i}\}_{i=1}^{n_0}$. For the first part in the right hand of Eq (13), we have:

$$
\begin{aligned}
&\sum_{i=1}^{n} -\log p(\boldsymbol{z}_i|y=y_i) \\
&\stackrel{c}{=} \frac{1}{2} \sum_{i=1}^{n} \|\boldsymbol{z}_i - \boldsymbol{\mu}_c^{\mathcal{D}}\|_2^2 \quad \triangleright p(\boldsymbol{z}|y=c) = \mathcal{N}(\boldsymbol{z}|\boldsymbol{\mu}_c^{\mathcal{D}}, \mathbf{I}) \\
&= \frac{1}{2} \sum_{c=1}^{C} \sum_{i=1}^{n_0} \|\boldsymbol{z}_{c_i} - \boldsymbol{\mu}_c^{\mathcal{D}}\|_2^2 \\
&= \frac{1}{2} \sum_{c=1}^{C} \sum_{i=1}^{n_0} \left\{ \|\boldsymbol{z}_{c_i}\|_2^2 - 2\boldsymbol{z}_{c_i}^{\mathrm{T}} \boldsymbol{\mu}_c^{\mathcal{D}} + \|\boldsymbol{\mu}_c^{\mathcal{D}}\|_2^2 \right\} \\
&\stackrel{c}{=} \frac{1}{2} \sum_{c=1}^{C} \left\{ \sum_{i=1}^{n_0} -2n_0\|\boldsymbol{\mu}_c^{\mathcal{D}}\|_2^2 + n_0\|\boldsymbol{\mu}_c^{\mathcal{D}}\|_2^2 \right\} \quad \triangleright \|\boldsymbol{z}\|_2^2 = 1 \\
&= \frac{1}{2} \sum_{c=1}^{C} \sum_{i=1}^{n_0} -n_0\|\boldsymbol{\mu}_c^{\mathcal{D}}\|_2^2 \\
&= \frac{1}{2} \sum_{c=1}^{C} \sum_{i=1}^{n_0} \sum_{j=1}^{n_0} -\frac{1}{n_0} \boldsymbol{z}_{c_i}^{\mathrm{T}} \boldsymbol{z}_{c_j} \\
&= -\sum_{i=1}^{n} \sum_{j:y_j=y_i} \frac{1}{2n_0} S_{ij}
\end{aligned}
\tag{14}
$$

Keep in mind that the mean of class c $\boldsymbol{\mu}_c^{\mathcal{D}} = \frac{1}{n_0} \sum_{i=1}^{n_0} \boldsymbol{z}_{c_i}$. For the second part in the right hand of Eq (13), we have:

$$
\begin{aligned}
&\sum_{i=1}^{n} \sum_{c=1, c \neq y_i}^{C} \log p(\boldsymbol{z}_i | y = c) \\
=& \frac{1}{2} \sum_{i=1}^{n} \sum_{c=1, c \neq y_i}^{C} -\|\boldsymbol{z}_i - \boldsymbol{\mu}_c^{\mathcal{D}}\|_2^2 \\
=& \frac{1}{2} \sum_{i=1}^{n} \sum_{c=1, c \neq y_i}^{C} \big\{ -\|\boldsymbol{z}_i\|_2^2 + 2\boldsymbol{z}_i^{\mathrm{T}} \boldsymbol{\mu}_c^{\mathcal{D}} - \|\boldsymbol{\mu}_c^{\mathcal{D}}\|_2^2 \big\} \\
\overset{c}{=}& \frac{1}{2} \sum_{i=1}^{n} \sum_{c=1, c \neq y_i}^{C} 2\boldsymbol{z}_i^{\mathrm{T}} \boldsymbol{\mu}_c^{\mathcal{D}} - (C-1)n_0 \sum_{c=1}^{C} \|\boldsymbol{\mu}_c^{\mathcal{D}}\|_2^2 \quad \rhd \|\boldsymbol{z}\|_2^2 = 1 \\
=& \frac{1}{2} \sum_{i=1}^{n} \sum_{j:y_j \neq y_i}^{C} \frac{2}{n_0} \boldsymbol{z}_i^{\mathrm{T}} \boldsymbol{z}_j - \frac{C-1}{n_0} \sum_{c=1}^{C} \sum_{i=1}^{n_0} \sum_{j=1}^{n_0} \boldsymbol{z}_{c_i}^{\mathrm{T}} \boldsymbol{z}_{c_j} \\
=& \sum_{i=1}^{n} \big\{ \sum_{j:y_j \neq y_i} \frac{1}{n_0} S_{ij} - \sum_{j:y_j = y_i} \frac{C-1}{2n_0} S_{ij} \big\}
\end{aligned}
\tag{15}
$$

According to Eq (14) and Eq (15), we have:

$$
\mathcal{L}_{Gen-Bin} \overset{c}{=} \sum_{i=1}^{n} \big\{ -\frac{1}{n_0} \sum_{j:y_j = y_i} S_{ij} + \frac{1}{(C-1)n_0} \sum_{j:y_j \neq y_i} S_{ij} \big\}
\tag{16}
$$

According to Eq (12) and Eq (16), we can obtain:

$$
\mathcal{L}_{MS} \overset{c}{\geq} \mathcal{L}_{Gen-Bin}
\tag{17}
$$

$\square$

# B  Details about Experiment Setup

## B.1  Split Datasets

In Table 6 we show the detailed statistics about Split datasets. On MNIST (LeCun et al., 1998), we randomly select 500 samples of the original training data for each class as training data stream, following previous works (Aljundi et al., 2019a;c; Jin et al., 2020). On CIFAR10 and CIFAR100 (Krizhevsky et al., 2009), We use the full training data from which 5% samples are regarded as validation set. The original miniImageNet dataset is used for meta learning (Vinyals et al., 2016) and 100 classes are divided into 64 classes, 16 classes, 20 classes respectively for meta-training, meta-validation and meta-test respectively. We merge all 100 classes to conduct class-incremental learning. There are 600 samples per class in miniImageNet. We divide 600 samples into 456 samples, 24 samples and 120 samples for training, validation and test respectively. We do not adopt any data augmentation strategy.

## B.2  Smooth Datasets

For a data stream with $C$ classes, we assume the length of stream is $N$ and $n_0 = \frac{N}{c}$. We denote $p_c(t)$ as the occurrence probability of class $c$ at time step $t$ and assume $p_c(t) \sim \mathcal{N}(t | \frac{(2c-1)n_0}{2}, \frac{n_0}{c})$. At each time step $t$, we calculate $p(t) = (p_1(t), ..., p_C(t))$ and normalize $p(t)$ as the parameters of a Categorical distribution from which a class index ct is sampled. Then we sample one data of class ct without replacement. In this setting, data distribution changes smoothly and there is no notion of task. We call such data streams as Smooth datasets.

Here we show the characteristics of Smooth datasets in detail. In Table 7, we list the detailed statistics of Smooth datasets. Due to the randomness in the data stream generation process, the number of

| Dataset | MNIST | CIFAR10 | CIFAR100 | miniImageNet |
|---|---|---|---|---|
| Image Size | (1,32,32) | (3,32,32) | (3,32,32) | (3,84,84) |
| #Classes | 10 | 10 | 100 | 100 |
| #Train Samples | 5000 | 47500 | 47500 | 45600 |
| #Valid Samples | 10000 | 2500 | 2500 | 2400 |
| #Test Samples | 10000 | 10000 | 10000 | 12000 |
| #Task | 5 | 5 | 10 or 20 | 10 or 20 |
| #Classes per Task | 2 | 2 | 10 or 5 | 10 or 5 |

Table 6: Details about Split datasets.

samples of each class is slightly imbalanced. On CIFAR100 and miniImageNet we set the mean number of samples of each class $n_0 = 400$ to avoid invalid sampling when exceeding the maximum number of sample of one class in the original training set. The last rows of Table 7 show the range of number of samples of each classes in our experiments. For example, in 15 repeated runs, 15 different Smooth CIFAR10 data streams are established. The number of samples of each class is in the interval $[464, 536]$. It should be emphasized that in all experiments we use the same random seed to obtain the identical 15 data streams for fair comparison.

Some existing works (Zeno et al., 2018; Lee et al., 2020) also experiment on task-free datasets where a data stream is still divided into different tasks but the switch of task is gradual instead of abrupt. In fact, in their settings, task switching only occurs in a part of time, so that in the left time the distribution of data streams is still i.i.d. In contrast, in our Smooth datasets the distribution changes at the class level which simulates task-free class-incremental setting. In addition, the distribution of data streams is never i.i.d, which can reflect real-world CL scenarios better.

In Figure 5, we plot the curves of training loss of ER-reservoir on Split CIFAR10 and Smooth CIFAR10. When a new task emerges (each task consists of 100 iterations), the loss on Split CIFAR10 increases dramatically. Thus some methods can detect the change of loss to inference the task boundaries although they are not informed and conduct additional offline training phase (Aljundi et al., 2019b). However, such a trick is not applicable on Smooth datasets therefore only the real online CL methods can work.

| Dataset | CIFAR10 | CIFAR100 | miniImageNet |
|---|---|---|---|
| #Classes ($C$) | 10 | 100 | 100 |
| #Train Samples (length of data stream) ($n$) | 5000 | 40000 | 40000 |
| #Valid Samples | 2500 | 2500 | 2400 |
| #Test Samples | 10000 | 10000 | 12000 |
| Mean number of samples of one class ($n_0$) | 500 | 400 | 400 |
| Minimum number of samples of one class | 464 | 357 | 354 |
| Maximum number of samples of one class | 536 | 450 | 446 |

Table 7: Details about Smooth datasets.

## B.3 HYPERPARAMETER SELECTION

In our method, although there are several hyperparameters in $\mathcal{L}_{MS}$ and $\mathcal{L}_{PNCA}$, we only need to tune $\gamma$ which is the weight of $\mathcal{L}_{PNCA}$ in $\mathcal{L}_{Hybrid}$. The value of other hyperparameters in $\mathcal{L}_{Hybrid}$ is fixed as stated in the main text. We select $\gamma$ from $[0, 0.1, 0.25, 0.5, 0.75, 1.0, 1.25, 1.5, 2.0]$. Especially, $\gamma = 0$ makes $\mathcal{L}_{Hybrid}$ become $\mathcal{L}_{MS}$.

We follow previous works to use SGD optimizer in all experiments. However, we find compared to CE loss, a wider range of learning rate $\eta$ should be searched in for DML losses. Previous works are based on CE loss and always set $\eta = 0.05$ or $\eta = 0.1$ (Aljundi et al., 2019c;a; Jin et al., 2020; Mai et al., 2021). We find the optimal $\eta$ for $\mathcal{L}_{Hybrid}$ is often larger than $0.1$. However, when a larger $\eta$ (e.g. $0.2$ on MNIST and CIFAR10) is used, baselines based on CE loss will degrade obviously because of the unstable results over multiple runs. Thus, for baselines we select $\eta$ from $[0.05, 0.1, 0.15, 0.2]$. For our

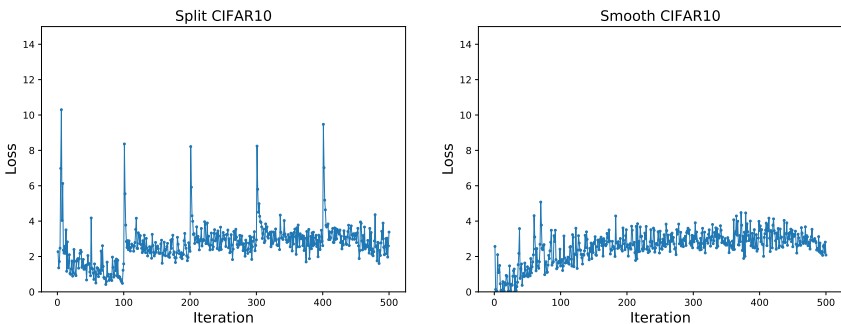

Figure 5: The curves of training loss on Split CIFAR10 (left) and Smooth CIFAR10 (right). Split CIFAR10 comprises of 5 tasks, 100 iterations per task.

method, we select $\eta$ from $[0.05, 0.1, 0.15, 0.2, 0.25, 0.3, 0.35, 0.4, 0.45, 0.5]$. The hyperparameters of our method are displayed in Table 8.

|  | Split Datasets | | | | Smooth Datasets | | |
|---|---|---|---|---|---|---|---|
|  | MNIST | CIFAR10 | CIFAR100 | miniImageNet | CIFAR10 | CIFAR100 | miniImageNet |
| $\gamma$ | 0.1 | 0.1 | 0.5 | 1 | 0.1 | 1.0 | 1.0 |
| $\eta$ | 0.05 | 0.2 | 0.35 | 0.2 | 0.25 | 0.5 | 0.2 |

Table 8: Hyperparamters of PNCA loss weight $\gamma$ and learning rate $\eta$ for our method on different datasets.

### B.4 EVALUATION METRICS

Following previous online CL works (Aljundi et al., 2019a; Mai et al., 2021), we use Average Accuracy and Average Forgetting (Chaudhry et al., 2019b) on Split datasets after training all $T$ tasks. Average Accuracy evaluates the overall performance and Average Forgetting measures how much learned knowledge has been forgotten. Let $a_{k,j}$ denote the accuracy on the held-out set of the $j$-th task after training the model on the first $k$ tasks. For a dataset comprised of $T$ tasks, the Average Accuracy is defined as follows:

$$A_T = \frac{1}{T} \sum_{j=1}^{T} a_{T,j} \tag{18}$$

and Average Forgetting can be written as:

$$f_j^T = \max_{l \in \{1,\dots,T-1\}} a_{l,j} - a_{T,j}, \forall j < T \tag{19}$$

$$F_T = \frac{1}{T-1} \sum_{j=1}^{T-1} f_j^T \tag{20}$$

We report Average Accuracy and Average Forgetting in the form of percentage. It should be noted that a higher Average Accuracy is better while a lower Average Forgetting is better.

On Smooth datasets, as there is no notion of task, we evaluate the performance with the accuracy on the whole held-out set after finishing training the whole training set.

### B.5 CODE DEPENDENCIES AND HARDWARE

The Python version is 3.7.6. We use the PyTorch deep learning library to implement our method and baselines. The version of PyTorch is 1.7.0. Other dependent libraries include Numpy (1.17.3), torchvision (0.4.1), matplotlib (3.1.2) and scikit-learn (0.22). The CUDA version is 10.2. We run all experiments on 1 NVIDIA RTX 2080ti GPU. We will publish our codes once the paper is accepted.

# C  MORE EXPERIMENTAL RESULTS

## C.1  EFFECT OF PROXY-NCA LOSS WEIGHT $\gamma$

In Table 6, we show the effect of Proxy-NCA loss weight $\gamma$ on split datasets. We can find on MNIST and CIFAR10 where the number of classes is relatively small, although the best results are obtained with $\gamma = 0.1$, we can have competitive results with $\gamma = 0$. In other words, only using MS loss is effective enough. However, on CIFAR100 and miniImageNet, as discussed in main text, the expected number of classes in each minibatch is larger which reduces the probability of occurrence of positive pair so that limits the learning ability of pair-based MS loss. At this time, it is necessary to introduce auxiliary PNCA loss. When $\gamma$ is larger than 1, the performance begins to degrade, which implies that excessively focusing on discriminative PNCA loss will affect the performance of the generative NCM classifier.

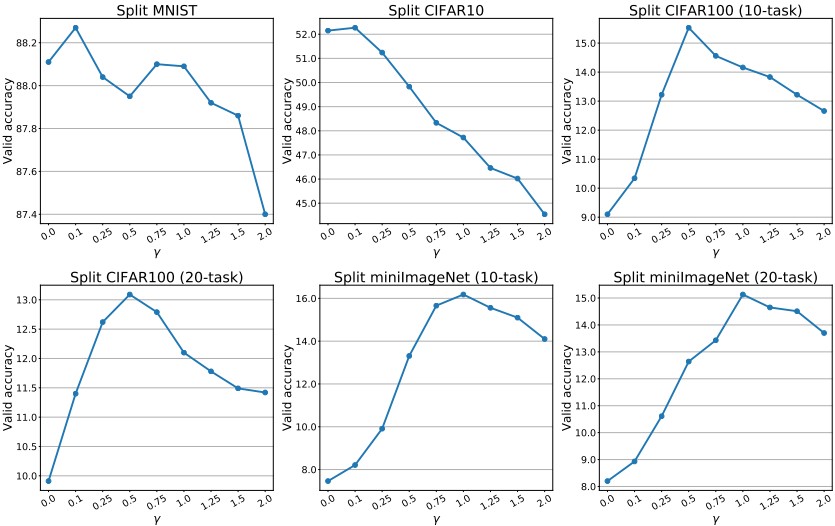

Figure 6: Average Accuracy on valid set of our method with different Proxy-NCA loss weight $\gamma$ in the 6 Split datasets as reported in Table 1 of main text. We report the mean of 15 runs.

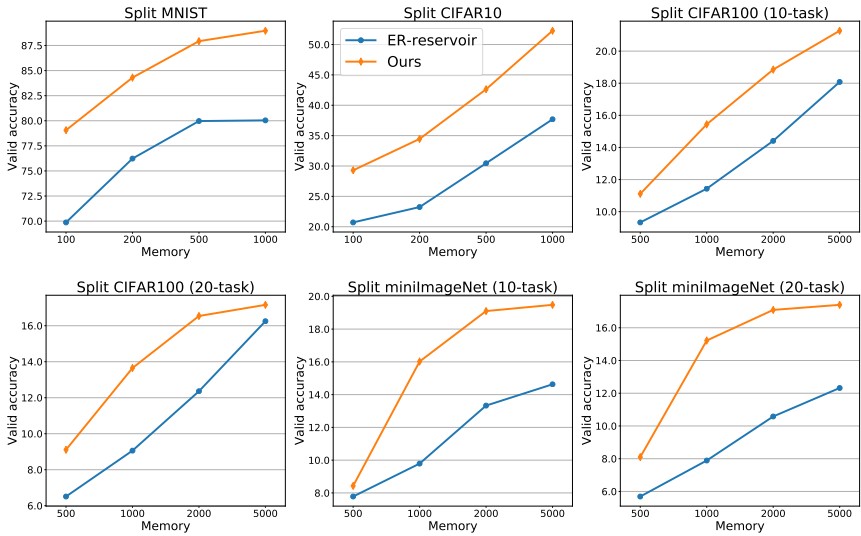

Figure 7: Average Accuracy on valid set of ER-reservoir and our method with different size of $\mathcal{M}$.

## C.2 Results with Different Memory Sizes

In Table 9 and Table 10, we report the performance of our method and all baselines compared in main text on Split CIFAR100 and miniImageNet with 2k memory size. Our method performs best in all settings and the improvements are obvious, which is similar with the results in 1k memory size settings reported in Table 1-2 of main text. We also report Average Accuracy of ER-reservoir and our method on valid set with a various of memory sizes. Our method outperforms ER-reservoir consistently which shows broad applicability of our method. The improvements are relatively small when memory size is 500 on CIFAR100 and miniImageNet, which is due to the fact that when size of replay memory $\mathcal{M}$ is too small (5 samples per class on average), class mean $\mu_c^{\mathcal{M}}$ cannot approximate the real class mean $\mu_c^{\mathcal{D}}$ well and thus proposed NCM classifier degrades.

## C.3 Comparison with Triplet Loss

Yu et al. (2020) proposes SDC method for conventional CIL based on NCM classifier and another pair-based DML loss, *triplet loss* (Hoffer & Ailon, 2015). As discussed in Related Work of main text, SDC is not applicable for online CIL. To further clarify our contribution given this work, we evaluate the performance of NCM classifier after training the model with the hybrid objective $\mathcal{L}_{Triplet} + \gamma \mathcal{L}_{PNCA}$, where $\mathcal{L}_{Triplet}$ represents triplet loss. Please see Hoffer & Ailon (2015) and (Yu et al., 2020) for details about $\mathcal{L}_{Triplet}$.

We report the results of the hybrid objective involving triplet loss and compare it with our hybrid loss ($\mathcal{L}_{MS} + \gamma \mathcal{L}_{PNCA}$) in Table 11. We can find our proposed loss is superior to $\mathcal{L}_{Triplet} + \gamma \mathcal{L}_{PNCA}$ consistently. In addition to replace softmax classifier with NCM classifier for online CIL problem, our contributions mainly reflect in introducing a hybrid of MS loss and PNCA loss in the view of training the generative NCM classifier. We believe the above results show MS loss is critical for online CIL, and introducing MS loss is not a trivial contribution, even if given the SDC work (Yu et al., 2020).

| Methods | CIFAR100 (10-task) | CIFAR100 (20-task) | miniImageNet (10-task) | miniImageNet (20-task) |
|---|---|---|---|---|
| fine-tune | 6.26±0.30 | 3.61±0.24 | 4.43±0.19 | 3.12±0.15 |
| ER-reservoir | 15.15±0.37 | 12.76±0.69 | 13.88±0.68 | 11.76±0.88 |
| A-GEM (Chaudhry et al., 2019a) | 6.50±0.16 | 3.61±0.08 | 4.41±0.14 | 3.23±0.13 |
| GSS-Greedy (Aljundi et al., 2019c) | 12.66±0.67 | 11.62±0.61 | 13.95±0.40 | 11.70±0.55 |
| MIR (Aljundi et al., 2019a) | 15.11±0.63 | 12.45±0.54 | 14.22±0.93 | 12.35±1.08 |
| GMED-ER (Jin et al., 2020) | 14.93±0.39 | 11.97±0.60 | 12.10±1.29 | 9.90±0.95 |
| GMED-MIR (Jin et al., 2020) | 15.11±0.50 | 12.01±0.81 | 13.90±0.58 | 12.25±0.59 |
| ASER$_\mu$ (Mai et al., 2021)[*] | 17.20±0.50 | – | 14.80±1.10 | – |
| Ours | **18.96±0.42** | **16.69±0.76** | **19.17±0.38** | **17.10±0.58** |
| i.i.d. online | 20.62±0.48 | 20.62±0.48 | 18.02±0.63 | 18.02±0.63 |
| i.i.d. offline | 45.59±0.29 | 45.59±0.29 | 38.63±0.59 | 38.63±0.59 |

Table 9: Average Accuracy of 15 runs on *Split* datasets. Higher is better. [*] indicates the results are from the original paper. The size of memory $\mathcal{M}$ is 2k.

| Methods | CIFAR100 (10-task) | CIFAR100 (20-task) | miniImageNet (10-task) | miniImageNet (20-task) |
|---|---|---|---|---|
| fine-tune | 51.60±0.77 | 65.51±0.78 | 41.12±0.82 | 52.99±0.89 |
| ER-reservoir | 40.19±0.71 | 52.45±0.85 | 32.91±0.79 | 44.93±0.81 |
| A-GEM (Chaudhry et al., 2019a) | 51.45±0.68 | 67.13±0.55 | 40.49±0.43 | 51.45±0.68 |
| GSS-Greedy (Aljundi et al., 2019c) | 40.02±0.81 | 45.90±2.09 | 30.50±0.83 | 39.09±2.34 |
| MIR (Aljundi et al., 2019a) | 39.82±0.79 | 48.54±0.60 | 29.27±1.32 | 39.74±1.50 |
| GMED-ER (Jin et al., 2020) | 42.99±0.75 | 55.45±0.68 | 31.81±1.73 | 43.68±1.23 |
| GMED-MIR (Jin et al., 2020) | 43.78±0.93 | 55.15±0.71 | 29.67±0.58 | 41.16±0.95 |
| $ASER_\mu$ (Mai et al., 2021)* | 38.60±0.60 | – | 22.20±1.60 | – |
| Ours | **26.59±0.58** | **29.53±0.94** | **17.82±0.36** | **24.10±0.71** |

Table 10: Average Forgetting of 15 runs on *Split* datasets. Lower is better. * indicates the results are from the original paper. The size of memory $\mathcal{M}$ is 2k.

| Loss | MNIST (5-task) | CIFAR10 (5-task) | CIFAR100 (10-task) | CIFAR100 (20-task) | miniImageNet (10-task) | miniImageNet (20-task) |
|---|---|---|---|---|---|---|
| $\mathcal{L}_{Triplet} + \gamma\mathcal{L}_{PNCA}$ | 87.06±0.52 | 43.56±1.63 | 13.21±0.28 | 10.93±0.34 | 14.31±0.33 | 13.55±0.22 |
| $\mathcal{L}_{MS} + \gamma\mathcal{L}_{PNCA}$ (Ours) | **88.79±0.26** | **51.84±0.91** | **15.56±0.39** | **13.65±0.35** | **16.05±0.38** | **15.15±0.36** |

Table 11: Comparison between MS loss and Triplet Loss. We report Average Accuracy of 15 runs on *Split* datasets. The size of memory $\mathcal{M}$ is 1k.

