# OpenReview forum: "Bypassing Logits Bias in Online Class-Incremental Learning with a Generative Framework"
_ICLR.cc/2022/Conference — ICLR 2022 Submitted_

### Official Review · Reviewer_aZyF · 2021-11-02

**Correctness:** 3
**Technical Novelty And Significance:** 3
**Empirical Novelty And Significance:** 3
**Recommendation:** 6
**Confidence:** 4

**Main Review:**

Pros:
•	The paper addresses an important and challenging problem, i.e., online continual learning.
•	The motivation is clear.
•	They conducted extensive experiments to show the consistent superiority of proposed method compared to existing state-of-the-art methods.
Cons:
•	The hypothesis of Proposition 1 is derived in the case of supervised learning and seems to have few connections with continual learning. Why the multi-similarity loss alleviates catastrophic forgetting in continual learning?
•	The hypothesis of this paper is based on that the feature is well discriminative. However, the feature extractor is usually underfitting in online setting. It is difficult to disentangle the effects of catastrophic forgetting from underfitting.
•	If the memory size is very small, the reserved samples of old classes are apt to be lost, how to obtain a generative classifier?
•	It is not suitable to train the model 5 epochs on i.i.d offline setting for almost datasets.


**Summary Of The Paper:**

This work develops a novel generative framework to bypass logits bias in online continual learning. They utilized a multi-similarity loss to weight feature pairs better during training time. At test time, they leveraged reserved samples to generate a generative classifier instead of a softmax classifier. Furthermore, they introduce an auxiliary loss to improve the ability of multi-similarity loss to learn from new data.

**Summary Of The Review:**

The motivition is clear, and experiments are comprehensive.
However, some unclear explanations and analyses are existed.

---

> ### Author Response · Authors · 2021-11-20
> **Response to Reviewer aZyF**
>
> Thanks for your insightful comments.
>
> * *“The hypothesis of Proposition 1 is derived in the case of supervised learning and seems to have few connections with continual learning. Why the multi-similarity loss alleviates catastrophic forgetting in continual learning?”*
>
> It is NCM classifier, not MS loss plays the principal role in alleviating catastrophic forgetting. We use MS loss because it can be explained as a generative optimization objection through Proposition 1, and thus suitable for the generative NCM classifier and boost performance.
>
> * *“The hypothesis of this paper is based on that the feature is well discriminative. However, the feature extractor is usually underfitting in online setting. It is difficult to disentangle the effects of catastrophic forgetting from underfitting.”*
>
> We indeed observe severe underfitting when using MS loss only. For this reason, we use the hybrid loss to improve the model’s ability to learn new knowledge.
>
> * *“If the memory size is very small, the reserved samples of old classes are apt to be lost, how to obtain a generative classifier?”*
>
> It is true that when the memory size is smaller than the number of classes, we cannot calculate class means. In this extreme circumstance, we can use proxies of PNCA loss as substitution of class means for the missing classes.
>
> * *“It is not suitable to train the model 5 epochs on i.i.d offline setting for almost datasets.”*
>
> We here follow the setting of previous works of online continual learning. i.i.d offline setting acts as not an absolute upper bound, but a reference substance to exhibit that the low data regime of online setting indeed leads to underfitting.

---

### Official Review · Reviewer_TwHY · 2021-11-02

**Correctness:** 3
**Technical Novelty And Significance:** 2
**Empirical Novelty And Significance:** 3
**Recommendation:** 5
**Confidence:** 4

**Main Review:**

Strengths:
-	Identifying generative classification as a promising alternative to softmax-based discriminative classification, and demonstrating that nearest-class mean classification (NCM) can be interpreted as generative classification.
-	Proposing a task-free solution for updating a feature extractor for NCM classification.
-	Proposing an interesting new benchmark for online continual learning that might be of interest to others working on this problem (i.e., the “smooth datasets”).
-	The reported performance of the proposed method is substantially better than the existing methods that are compared against.


Weaknesses:
-	My main issue with this paper is that its two main contributions (i.e., [1] proposing generative classification for class-incremental learning and [2] proposing an online variant of NCM classification for class-incremental learning) have been recently proposed by two other papers, but this paper does not discuss, compare against or cite either of them. Specifically, Van De Ven et al. (2021, CVPR-W; https://arxiv.org/abs/2104.10093) argued for addressing class-incremental learning using generative classification, and De Lange & Tuytelaars (2021, ICCV; https://arxiv.org/abs/2009.00919) proposed an online version of an NCM-based class-incremental learning method.
-	Another relevant paper that would have been good to discuss (and compare against) is Lomonaco et al. (2020, CVPR-W; https://arxiv.org/abs/1907.03799), which also proposes an online class-incremental learning method that addresses the logits bias problem.
-	It is unclear to me exactly what the authors mean by “online continual learning”. Does it mean that there are no clear task boundaries, or does it mean that task boundary information is not provided to the network? At the top of section 4.1, it is stated that “task boundaries are not informed during training”. However, it seems to me that because class labels are provided, task boundaries can trivially be derived? It is also unclear to me why task-incremental learning cannot be performed in the online setting? (As claimed in the introduction.)
-	I’m not sure how relevant the motivation of generative classification as a more efficient method in the low data regime is for this paper. My feeling is that for all problems considered in this paper, if the data is provided to the algorithm in an i.i.d. stream, the discriminative classifier would perform better than the generative classifier.


Minor comments:
-	An accuracy of 15% on CIFAR-100 or miniImageNet, although perhaps better than the compared methods, is not of any practical value for real-world applications. Perhaps this performance improvement is of interest from an academic perspective, but it does not seem to be of value for any practical applications. I think it would be good to discuss this.
-	What is meant by “virtual catastrophic forgetting”? It would be good to expand a bit on that.


**Summary Of The Paper:**

To address the problem of logits bias in class-incremental learning with deep neural networks, this paper proposes to replace the typically used softmax output layer by a nearest-mean classifier (on the feature space), which can be interpreted as a generative classifier. To train the feature attractor in an online setting (i.e., without reliance on task boundaries), the paper proposes a hybrid loss consisting of a multi-similarity loss (which can be interpreted as an approximate generative objective) and an auxiliary “proxy” loss. The paper reports substantial performance improvements on a number of computer vision benchmark (including ones with natural images) over a selection of existing methods.

Additionally, the paper proposes a new “smooth dataset” benchmark for online class-incremental learning, whereby the transitions between classes are smooth rather than based on abrupt task transitions.


**Summary Of The Review:**

As explained above, my main issue with this paper is that its two main contributions are not as novel as suggested in the paper, as similar contributions were recently made by three other papers (https://arxiv.org/abs/2009.00919 and https://arxiv.org/abs/2104.10093 and https://arxiv.org/abs/1907.03799). There are some differences with these papers, but I think it is necessary that this paper clearly discusses what it contributes on top of these papers. When that is done, it should be better possible to judge the additional contribution of this paper.

My current scores are my “expectations” for this paper after a discussion of the above papers (and corresponding moderation of some novelty claims) has been incorporated. If the authors can make a more convincing case that their paper makes important contributions on top of those made by these previous papers, I’d be happy to increase my score.

---

> ### Author Response · Authors · 2021-11-20
> **Response to Reviewer TwHY**
>
> Thanks for your valuable comments.
>
> * *Comparison with Van De Ven et al.’s and Lomonaco et al.’s methods.*
>
> Since these two methods need a pretrained model as either a fixed feature extractor or initialization parameters, they somewhat simplify the problem. For this reason, it’s unable to compare our method with them fairly.
>
> * *Comparison with De Lange & Tuytelaars’s CoPE method.*
>
> Thanks for your suggestion. We have added comparison with CoPE in the new revision. The results show that our method outperforms CoPE in terms of accuracy, forgetting and efficiency.
>
> * *About the definition of “online continual learning”.*
>
> We follow the setting used in MIR and ASER. The differences between “online continual learning” and “continual learning” are:
>
> 1. The model does not have any information about tasks, including task lengths, boundaries, identifiers etc. So, the task boundaries can either “do not exist” or “exist are not provided to model” (corresponding to Smooth datasets and Split datasets respectively). This also explains why task-incremental learning cannot be performed in the online setting: because in task-incremental learning, task identifiers are provided to model in both training and inference stage.
>
> 2. Each training data passes the model only once. This is a consequence of 1., since if the model trained a task for several epoch, the model must have stored all samples of current tasks and know the task boundaries.
>
> Therefore, although the class labels are provided to the model, the model cannot identify which task the class belongs to (or even no task exists as in Smooth datasets).
>
> * *“if the data is provided to the algorithm in an i.i.d. stream, the discriminative classifier would perform better than the generative classifier.”*
>
> It’s true. However, in the online CIL scenario, ER-based methods store a very small number of samples for each class (10 samples per class in Split CIFAR-100). Through the replay, the discriminative classifier would overfit these stored samples and thus perform worse. Studies about few-shot learning (such as [1]) also encounter this problem and thus choose to use a generative classifier.
>
> * *About practical applications of our method*
>
> Split CIFAR100 and Split MiniImagenet are two difficult datasets in the field of online continual learning, and all existing online CIL methods perform poorly on them. If we want to improve the performance of online CIL methods in practice, a straightforward method is to use a pretrain model like Van De Ven et al.’s and Lomonaco et al.’s methods (although the use of pretraining models is not the mainstream of online CIL research currently).
>
> * *About “virtual catastrophic forgetting”*
>
> I’m sorry to use a confusing expression here. We have replaced “virtually” with “actually” in the revision.
>
> [1] Jake Snell, Kevin Swersky, Richard S. Zemel. Prototypical networks for few-shot learning. NIPS 2017.

---

> > ### Comment · Reviewer_TwHY · 2021-11-22
> > **Response to first Author Rebuttal**
> >
> > Thank you for your response.
> >
> > I think the way the authors describe “online continual learning” in the rebuttal is a lot clearer than the way they describe it in the paper itself. I encourage the authors to incorporate the description from their rebuttal into the paper itself.
> >
> > I’m afraid the authors have not addressed my main concerns.
> >
> > As indicated in my original review, the two main contributions / conceptual ideas that are put forward by this paper were recently proposed by two other papers (i.e., Van De Ven et al., 2021 proposed generative classification for class-incremental learning and De Lange & Tuytelaars, 2021 proposed an online version of NCM). In addition, an important claim of this paper is that they are the first to address the logits bias in online continual learning, but I pointed the authors to an earlier paper that did so (Lomonaco et al., 2020).
> >
> > The authors declined to discuss the relation of their work with Van De Ven et al. (2021) and Lomonaco et al. (2020), giving as reason that these papers used pre-training.
> > Firstly, my main concern was not about performing a direct comparison with these papers, but it was about discussing how the theoretical / conceptual advance of the current paper relates to the theoretical / conceptual ideas from these pervious papers. This is especially important because the authors claim to make certain novel contributions, which were also made by these papers. I wanted to give the authors a chance to make an argument why their paper still makes an important contribution on top of these previous papers (i.e., by clearly discussing the relation of their work to this pervious work), but unfortunately the authors did not (yet?) do this.
> > Secondly, the argument of pre-training is at best problematic. The methods proposed by both of these papers do not require pre-training. They can perfectly well be run without pre-training, and Van De Ven et al. (2021) even report a number of experiments without pre-training.
> >
> > In the revised manuscript, the authors did include a comparison with the CoPE method proposed by De Lange & Tuytelaars (2021), although they still do not discuss how the conceptual advance of their paper relates to the conceptual advance of this paper. Moreover, the newly reported results for CoPE are reason for concern. The performances reported by the authors are substantially lower than the performances reported in the original paper. Could the authors explain this difference? As far as I could see, the authors do not provide any details about how they run the CoPE method.
> >
> > Finally, I think from the author’s response it is clear that the motivation of generative classification as a more efficient method in the low data regime is not relevant for this paper. In the rebuttal, the authors now instead make an argument against the use of experience replay. But I think this argument misses the point, because (1) it doesn’t explain why generative classification is better for class-incremental learning than discriminative classification, and (2) the current paper also uses experience replay.

---

### Official Review · Reviewer_6SSs · 2021-11-03

**Correctness:** 3
**Technical Novelty And Significance:** 2
**Empirical Novelty And Significance:** 2
**Recommendation:** 5
**Confidence:** 4

**Main Review:**

**Strengths**
- S1: Providing a theoretical argument that the minimizing MS loss is optimizing a generative model with the assumption of uniform distribution of $p(y)$.
- S2: Computational overhead due to the proposed method is negligible thus practically useful.
- S3: Pretty compelling accuracy gain in CIFAR10 (5-task), CIFAR-100 (20-task), miniImageNet (10/20-task).

**Weaknesses**
- W1: The proposed method is a combination of existing loss. The novelty of technical contribution is not very strong.
- W2: The proposed hybrid loss is argued that it is beneficial as the Proxy-NCA loss will promotes learning new knowledge better (first paragraph in Sec. 3.4), rather than less catastrophic forgetting. But the empirical results show that the proposed method exhibits much less forgetting than the prior arts (Table. 2). The argument and the empirical results are not well aligned.
	- Also, as the proposed method seems promoting learning new knowledge, it is suggested to empirically validate the benefit of the proposed approach by a measure to evaluate the ability to learn new knowledge (e.g., intransigence (Chaudhry et al., 2018)).
- W3: Missing important comparison to Ahn et al.'s method in Table 3 (and corresponding section, titled "comparison with Logits Bias Solutions for Conventional CIL setting").
- W4: Missing analyses of ablated models (Table 4). The proposed hybrid loss exhibits meaningful empirical gains only in CIFAR100 (and marginal gain in CIFAR10), comparing "MS loss with NCM (Gen)" and "Hybrid loss with NCM (Gen)". But there is no descriptive analysis for it.
- W5: Lack of details of *Smooth* datasets in Sec. 4.3.
- W6: Missing some citation (or comparison) using logit bias correction in addition to Wu et al., 2019 and Anh et al., 2020
	- Kang et al., 2020: https://ieeexplore.ieee.org/stamp/stamp.jsp?tp=&arnumber=9133417
	- Mittal et al., 2021: https://openaccess.thecvf.com/content/CVPR2021W/CLVision/papers/Mittal_Essentials_for_Class_Incremental_Learning_CVPRW_2021_paper.pdf
- W7: Unclear arguments or arguments lack of supporting facts
	- 4th para in Sec.1
		- 'It should be noticed that in online CIL setting the data is seen only once, not fully trained, so it is analogous to the low data regime in which the generative classifier is preferable.'
			- Why?
	- 5th line of 2nd para in Sec. 3.1.3
		- 'This problem becomes more severe as the the number of classes increases.'
			- Lack of supporting facts
- W8: Some mistakes in text (see details in notes below) and unclear presentations

**Note**
- Mistakes in text
	- End of 1st para in Sec.1: intelligence agents -> intelligent agents
	- 3th line of 1 para in Sec. 3.2: we can inference with -> we can conduct inference with
	- 1st line of 1st para in Sec. 3.3
		- we choose MS loss as training... -> we choose the MS loss as a training...
		- MS loss is a state-of... -> MS loss is the state-of...
	- 6th line of Proposition 1: 'up to an additive constant and L' -> 'up to an additive constant c and L'
- Improvement ideas in presentations
	- Fig. 1: texts in legends and axis labels should be larger
	- At the beginning of page 6: Proposition (1) -> Proposition 1. --> (1) is confused with Equation 1.
	- Captions and legend's font should be larger (similar to text size) in Fig. 2 and 3.

**Summary Of The Paper:**

The paper proposes a hybrid loss of generative classification by NCM as a form of Multi-Similarity (MS) loss (Wang et al., 2019) and discriminative classification by Proxy-NCA (Movshovits-Attias et al., 2017). The paper contains a theoretical argument that the MS loss upper-bounds the 'Gen-Bin loss', which is a generative classification model. The hybrid loss shows decent gains in 'average accuracy' (Table 1) and large gain in 'forgetting' (Table 2) in the experiments with multiple datasets including MNIST, CIFAR-10/100, miniImageNet with smaller computational complexity than the prior arts (Fig. 4).

**Summary Of The Review:**

Despite the benefits of the theoretical arguments and somewhat convincing results, lack of novelty and missing details prevents this paper from being a quality to be accepted now.

---

> ### Author Response · Authors · 2021-11-20
> **Response to Reviewer 6SSs**
>
> Thanks for your detailed review and valuable comments.
>
>  * *“The proposed hybrid loss is argued that it is beneficial as the Proxy-NCA loss will promotes learning new knowledge better, rather than less catastrophic forgetting. But the empirical results show that the proposed method exhibits much less forgetting than the prior arts.”*
>
> It’s true. The results indeed show that hybrid loss forgets less than other baselines, but this does not contradict the statement that PNCA loss promotes the learning of new knowledge on the basis of MS loss. In our experiments, we empirically find that training the model only with MS loss on Split-CIFAR100 or Split-MiniImagenet can achieve both low accuracy and low forgetting. We attribute this phenomenon to the poor ability of MS loss to learn new tasks, since there are few positive pairs available in the current batch in Split-CIFAR100 or Split-MiniImagenet. For this reason, we introduce a discriminative PNCA loss whose protypes act as “anchors” to provide positive pairs stably.
>
> * *Missing comparison with Ahn et al.’s, Kang et al.’s, and Mittal et al.’s methods.*
>
> These works focus on class-incremental learning in which explicit task boundaries are informed to the model. As they depend on task boundaries which are not available in online CIL, these solutions cannot be applicable for online CIL directly unless nontrivial modifications are applied. Nevertheless, BiC and iCaRL are two methods that can be easily modified to satisfy the online CIL scenario, so we report their performance in Table 3.
>
> * *Missing analyses of ablated models.*
>
> We have now added related statements in “Ablation Study” section, and we would like to elaborate on it here. As we state in Sec. 3.4, we introduce PNCA auxiliary loss into our work because of the poor ability of MS loss to learn new tasks when there are too many classes in a single coming batch (as in Split-CIFAR100 and Split-MiniImagenet). While in CIFAR-10 which has only two classes in the new batch, the positive pair is enough, so it is expected that the help of PNCA loss is not significant.
>
> * *Lack of details of Smooth datasets in Sec. 4.3.*
>
> We have added details of Smooth datasets in Appendix Sec. B.2.
>
> * *About unclear arguments or arguments lack of supporting facts.*
>
> 1. *"It should be noticed that in online CIL setting the data is seen only once, not fully trained, so it is analogous to the low data regime in which the generative classifier is preferable.*
>
> In the online CIL scenario, each sample is seen by the model only once, leading to the underfitting of the model, so it is analogous to the low data regime. Table 1 shows an intuitive result that performances of i.i.d. online are much worse than i.i.d. offline, which manifests one-pass training indeed leads to underfitting of the model. Then, according to the conclusions of Ng & Jordan [1], Yogatama et al. [2] and Ding et al. [3], generative classifier performs better in this scenario.
>
> 2. *"This problem becomes more severe as the the number of classes increases."*
>
> Thank you for pointing it out. We have deleted this imprecise statement.
>
> * *Mistakes in text and unclear presentations*
>
> Thanks a lot for your suggestions. We have corrected the mistakes and updated the figures.
>
> [1] On discriminative vs. generative classifiers: A comparison of logistic regression and naive bayes. NIPS 2001.
>
> [2] Generative and discriminative text classification with recurrent neural networks. CoRR abs/1703.01898.
>
> [3] Discriminativelytuned generative classifiers for robust natural language inference. EMNLP 2020.

---

> > ### Comment · Reviewer_6SSs · 2021-11-28
> > **Response to the authors' response**
> >
> > Thank you for the detailed comments to my questions. I have the following concerns still persisting.
> > - **Learning new knowledge better**: Would you be able to present the intransigence as evidence?
> > -  **Comparison to offline CL methods (Ahn et al.)**: What is the nontrivial modification? Is it simply training once?
> > - **Unclear argument** *"It should be noticed that in online CIL setting the data is seen only once, not fully trained, so it is analogous to the low data regime in which the generative classifier is preferable.*: My point was how to learn a generative classifier under the low data regime. Adversarial training could be one solution but it's not the magic bullet. Given that the generative model is not widely deployed in few-shot learning literature, this argument may not be true. (The referred citations are not for low data regime, I believe. Correct me if I'm wrong)
> > - **Presentation improvements**: I still see that the figures are not changed. Did you update the figure?

---

### Official Review · Reviewer_hxsn · 2021-11-03

**Correctness:** 3
**Technical Novelty And Significance:** 2
**Empirical Novelty And Significance:** 2
**Recommendation:** 5
**Confidence:** 4

**Main Review:**

The paper is somewhat original and the text is relatively clear.

The authors describe the proposed approach in detail, and they present plenty of experimental results. The experimental section in general has convinced me that the proposed approach does indeed outperform the state of the art.

The authors argue that their method outperforms the state of the art because it optimizes the feature space in a generative way and thus enjoys the benefits of using a generative classifier. In combination with the NCM classifier, the authors propose a hybrid generative-discriminative loss.

The authors show that the NCM classifier can be interpreted as a very rudimentary generative model (i.e., an isometric Gaussian). This makes sense but it does not justify using a NCM classifier instead of a softmax. Moreover, the authors are never in this paper involved in actually generating latent-space data. Finally, the NCM classifier can also be interpreted as a discriminative model as it divides the latent space into Voronoi cells, hence modelling the decision boundaries between the different classes.

Regarding the experimental results, the authors should probably compare their approach to the SCR method as well (see [1]).

In my view, the major downside of this paper is that it looks like an exercise in achieving better numbers than the baselines, instead of an attempt to document and transmit the knowledge gained from the underlying study. Having read the paper twice, I cannot say that I learned something new, apart from the fact that the proposed method outperforms the state of the art. In my opinion, the explanation provided as to why that is is only speculation. A paper like this needs to describe in detail assumptions about the specific reasons leading to superior performance and test whether these assumptions hold water. In short, I think this paper will benefit substantially by attempting to discover the underlying factors behind the improved performance.

\
Questions
- If deep-metric-learning losses and the cross-entropy both maximize the mutual information between features and labels why would you expect different results?
- Can you explain what do you mean by stating that the MS loss optimizes the feature space in a generative way?
- What is the motivation behind selecting only 5k training instance for split MNIST? The paper you referenced cites two papers none of which uses the split MNIST benchmark.
- By comparing rows 3 and 6 of Table 4 it seems to me that using the hybrid loss instead of only the MS loss is only beneficial in the case of CIFAR-100. Why is that?

\
[1] Mai, Z., Li, R., Kim, H., & Sanner, S. (2021). Supervised Contrastive Replay: Revisiting the Nearest Class Mean Classifier in Online Class-Incremental Continual Learning. In Workshop on Continual Learning in Computer Vision at Conference on Computer Vision and Pattern Recognition (CVPR), 2021

**Summary Of The Paper:**

The authors attempt to improve online continual learning (CL) performance by eliminating the logit bias of the classifier used. They use a nearest-class-mean (NCM) classifier and a multi-similarity metric learning loss coupled with an auxiliary loss to achieve a good plasticity-stability balance.

**Summary Of The Review:**

In my view, the paper fails to adequately explain why the proposed method achieves the results it does, and there is a relevant baseline missing. Hence, I recommend rejection.

---

> ### Author Response · Authors · 2021-11-20
> **Response to Reviewer hxsn**
>
> Thanks for your insightful comments.
>
> * *Comparison with SCR*
>
> SCR has outstanding performance, but it is highly dependent on data augmentation (DA). For a fair comparison, we remove the DA operation of SCR and report its results on Split CIFAR10 and 10-task Split CIFAR100 with memory size of 1k. The results show that our method outperforms SCR.
>
> > &nbsp;&nbsp;&nbsp;&nbsp;&nbsp;&nbsp;&nbsp;&nbsp;&nbsp;&nbsp;&nbsp;&nbsp;&nbsp;&nbsp;&nbsp;&nbsp;&nbsp;&nbsp;&nbsp;&nbsp;&nbsp;&nbsp;&nbsp;&nbsp;&nbsp; Split CIFAR10   &nbsp;&nbsp;&nbsp;&nbsp;  Split CIFAR100
>
> >SCR w/o DA &nbsp;&nbsp;&nbsp;&nbsp;&nbsp;&nbsp;&nbsp;&nbsp;&nbsp;&nbsp; 44.03 &nbsp;&nbsp;&nbsp;&nbsp;&nbsp;&nbsp;&nbsp;&nbsp;&nbsp;&nbsp;&nbsp;&nbsp;&nbsp;&nbsp;&nbsp;&nbsp;&nbsp; 8.91
>
> >Ours&nbsp;&nbsp;&nbsp;&nbsp;&nbsp;&nbsp;&nbsp;&nbsp;&nbsp;&nbsp;&nbsp;&nbsp;&nbsp;&nbsp;&nbsp;&nbsp;&nbsp;&nbsp;&nbsp;&nbsp;&nbsp;&nbsp;&nbsp;&nbsp;**51.84**&nbsp;&nbsp;&nbsp;&nbsp;&nbsp;&nbsp;&nbsp;&nbsp;&nbsp;&nbsp;&nbsp;&nbsp;&nbsp;&nbsp;&nbsp;&nbsp;&nbsp;&nbsp;**15.56**
>
> * *“Can you explain what do you mean by stating that the MS loss optimizes the feature space in a generative way?”*
>
> As Boudiaf et al. [1], the mutual information $I(Z, Y)$ can be decomposed from two views: discriminative view $H(Y)-H(Y|Z)$ and generative view $H(Z)-H(Z|Y)$. We say MS loss optimizes the feature space in a generative way, because Boudiaf et al. show that MS loss modeling $H(Z)$ and $H(Z|Y)$ respectively, maximizing the former and minimizing the latter. Boudiaf et al. also indicate that CE loss optimizes features in a discriminative way, because it directly models and minimizes $H(Y|Z)$.
>
> * *“If deep-metric-learning losses and the cross-entropy both maximize the mutual information between features and labels why would you expect different results?”*
>
> Intuitively, when optimizing the feature extractor with MS loss which models $H(Z)$ and $H(Z|Y)$, it may have a better grasp of the probability distribution of $Z$ and $Z|Y$, and hence benefits a generative NCM classifier. Then in Proposition 1, we theoretically prove that MS loss maximizes the log-likelihood of class-conditional generative classifiers (with a regularizer against feature collapsing), which can be regarded as an MLE process of features distributions. With MS loss, the training and inference stages are unified into a generative framework.
>
> * *“What is the motivation behind selecting only 5k training instance for split MNIST? The paper you referenced cites two papers none of which uses the split MNIST benchmark.”*
>
> Actually, we follow the setting used in MIR and GMED for split MNIST. When using the whole training set for split MNIST, the performances of the different methods are very similar. We believe using the more difficult setting with only 5k training samples for split MNIST can better demonstrate the ability of different methods.
>
> * *“By comparing rows 3 and 6 of Table 4 it seems to me that using the hybrid loss instead of only the MS loss is only beneficial in the case of CIFAR-100. Why is that?”*
>
> We would like to clarify our motivation to use a hybrid loss. We empirically found if only trained with MS loss, the model’s performance degenerates as the expected number of classes in the coming new batch at each iteration increases. This phenomenon is attributed to the inadequate ability to learn from new data when there are few positive pairs available in the new batch. For instance, in 10-task split CIFAR100 with a batch size of 10, the expected number of instances form each class in a new batch is 1, which means the model can hardly learn to pull features from the same class closer. To remedy this problem, we introduce a discriminative PNCA loss whose protypes act as “anchors” to provide positive pairs stably. In CIFAR-10 which has only two classes in the new batch, the positive pair is enough, so the help of PNCA loss is not significant.
>
>
> [1] Malik Boudiaf, Jérôme Rony, Imtiaz Masud Ziko, Eric Granger, Marco Pedersoli, Pablo Piantanida, and Ismail Ben Ayed. A unifying mutual information view of metric learning: Cross-entropy vs. pairwise losses. In ECCV, 2020.

---

> > ### Comment · Reviewer_hxsn · 2021-11-29
> > **Reply to Rebuttal**
> >
> > My main issue with the paper (lack of clarity regarding where the increased learning performance is coming from) has not been resolved. I would advise the authors to investigate experimentally why the proposed approach outperforms the others. I acknowledge the authors' theoretical arguments, but, in my view, they are vague and speculative.
> >
> > I would also like to see a more information on how the NCM is used in practice, and, in particular, how the class means are maintained. For instance, are you computing the class means from scratch before every update step or are you using running averages, or something else? (In all fairness, I did not make this remark in my original rebuttal but I think the paper would be more complete if the relevant discussion is added.)
> >
> > The authors partially answered my questions. I disagree with the answer on MNIST-5k as I know from personal experience that this benchmark is not necessarily more difficult, but some methods tend to underfit the data because the training consists of only 500 updates. Hence, in my opinion, it does not make sense to set a restriction to use each mini-batch only once in online continual learning, especially when taking into account the vast differences in computational complexity between different methods. Even if you perform two or more training steps with each incoming mini-batch the learning is still online since the model is trained having access only to very small batches of data (and not all the data from the current task).

---

### Decision · Program_Chairs · 2022-01-20

**Decision:**

Reject

**Comment:**

This paper presents an approach for online continual learning where only a single pass over each task's data is allowed. Instead of the oft-used softmax classification setting in continual learning, the paper proposes to use the generative setting based on the nearest class mean (NCM). The paper claims that it avoids the logits bias problem in the softmax classifier and helps combat catastrophic forgetting.

While the reviewers found the basic idea interesting, there were concerns about novelty and lack of clarity regarding the reasons for improved performance. In particular, there are several aspects from existing work that are leveraged in this paper (e.g, replay, metric learning loss, combination of generative and discriminative classification, etc) but the paper lacks in establishing which of these components affect the performance and in what ways.

The authors and reviewers engaged in detailed discussions; however, the reviewers were still unsatisfied and did not change their assessment. Based on my own reading of the paper as well as going through the reviews and discussions, I too concur with their assessment. It would be a stronger paper if the paper could shed more light on the above aspects as well as address the other concerns raised by the reviewers. However, in the current shape, it is not ready for publication.